# Effect of Various Peening Methods on the Fatigue Properties of Titanium Alloy Ti6Al4V Manufactured by Direct Metal Laser Sintering and Electron Beam Melting

**DOI:** 10.3390/ma13102216

**Published:** 2020-05-12

**Authors:** Hitoshi Soyama, Fumio Takeo

**Affiliations:** 1Department of Finemechanics, Tohoku University, Sendai 980-8579, Japan; 2Department of Industrial Systems Engineering, National Institute of Technology, Hachinohe College, 16-1 Uwanotai, Tamonoki, Hachinohe 039-1192, Japan; takeo-m@hachinohe-ct.ac.jp

**Keywords:** additive manufacturing, post-processing, fatigue, Ti6Al4V, cavitation peening, laser peening, shot peening, direct metal laser sintering, electron beam melting

## Abstract

Titanium alloy Ti6Al4V manufactured by additive manufacturing (AM) is an attractive material, but the fatigue strength of AM Ti6Al4V is remarkably weak. Thus, post-processing is very important. Shot peening can improve the fatigue strength of metallic materials, and novel peening methods, such as cavitation peening and laser peening, have been developed. In the present paper, to demonstrate an improvement of the fatigue strength of AM Ti6Al4V, Ti6Al4V manufactured by direct metal laser sintering (DMLS) and electron beam melting (EBM) was treated by cavitation peening, laser peening, and shot peening, then tested by a plane bending fatigue test. To clarify the mechanism of the improvement of the fatigue strength of AM Ti6Al4V, the surface roughness, residual stress, and surface hardness were measured, and the surfaces with and without peening were also observed using a scanning electron microscope. It was revealed that the fatigue strength at *N* = 10^7^ of Ti6Al4V manufactured by DMLS was slightly better than that of Ti6Al4V manufactured by EBM, and the fatigue strength of both the DMLS and EBM specimens was improved by about two times through cavitation peening, compared with the as-built ones. An experimental formula to estimate fatigue strength from the mechanical properties of a surface was proposed.

## 1. Introduction

Additive manufactured (AM) titanium alloy is an attractive material for medical implants [1,2,3,4] and aviation components [5,6,7], as the geometry of the parts are directly produced from computer-aided design (CAD) data, and the lead time for production can be reduced remarkably. However, the fatigue strength of AM titanium alloy is significantly weaker, compared to that of wrought materials [5,6,8,9,10,11]. As manufacturing conditions and post-processing affect fatigue properties [5,6,12], and mechanical surface treatment improves fatigue strength [5,13,14,15], it is worthwhile to develop peening methods to improve the fatigue strength of AM titanium alloy.

As is well known, in AM metallic materials, tensile and compressive residual stress are introduced during the AM process [6,16] as a heat source is deposited locally on the material surface; this produces a three-dimensional distribution of temperature in the AM process which is similar to welding [17,18]. Moreover, residual stress affects fatigue performance [19,20]. Thus, the relaxation of residual stress improves the fatigue life and fatigue strength of AM titanium [21]. A heat treatment after the AM process is one way to improve fatigue properties [21,22]. As defects and pores are also produced in AM metallic materials, hot isostatic pressing (HIP) can improve the fatigue properties of AM titanium alloy [22,23,24,25,26,27,28]. The surface roughness caused by un-melted particles is one of the reasons for the weak fatigue performance [29,30]. However, HIP cannot improve fatigue performance, which is closely affected by surface defects, and it is difficult to apply the HIP process with large-scale components. It was reported that, in the case of AM titanium with an as-built surface, the fatigue life depended on the crack initiation from the rough surface, compared with internal defects [31]. The combined treatment of HIP and chemical etching was reported to improve fatigue properties [32]. There have been a lot of reports that revealed an improvement of fatigue properties by mechanical finishing or polishing after HIP [8,10,11,21,24,25,27,31,33,34,35,36]. However, some shapes cannot be machined, and the benefits of the topology optimization of AM metals [5,37] are lost.

As mentioned above, a large surface roughness was introduced by the AM process [29,30], cracks were initiated on the as-built surface of AM Ti6Al4V, compared with the machined surface [38], and the relation between the surface roughness and fatigue life of AM Ti6Al4V was reported [39]. Regarding the effect of surface roughness on AM Ti6Al4V, the Kitagawa–Takahashi plot [40], El Haddad–Topper approach [41], and root area parameter [42] have been investigated [43,44]. It was also reported that the fatigue life of AM Ti6Al4V was estimated by the equivalent initial flaw size (EIFS) [45]. In order to reduce the surface roughness, without machining, to improve the fatigue properties of AM Ti6Al4V, surface finishing, such as tribofinishing [46], vibratory finishing [47], electropolishing [48], tumbling [49], barrel polishing [50], chemically accelerated vibratory finishing [51], and cavitation abrasive finishing [15,52] have been proposed. Laser polishing of AM Ti6Al4V was also proposed [53,54]; however, the effect of laser polishing on fatigue properties was not investigated. As is well known, large surface irregularities, such as in as-built AM metals, can be smoothed, as shot peening can deform metallic surfaces. It was reported that the fatigue strength of AM Ti6Al4V at 10^7^ cycles was in the order of shot peening, HIP, tribofinishing, electropolishing, and as-built surfaces [46]. Thus, peening methods are some of most effective tools for improving the fatigue properties of AM Ti6Al4V.

While there are several methods for AM Ti6Al4V, the two major methods are direct metal laser sintering (DMLS) and electron beam melting (EBM). In both cases, the environment, temperature, and material particles during the AM process are different. When the fatigue strength of the DMLS and EBM specimens was compared, it was also found that the DMLS specimen had a low roughness and high fatigue strength [31,34,55,56,57,58,59,60,61], and the DMLS and EBM specimens were similar [21]. It was also reported that DMLS and EBM specimens have different metallographic structures [61], and the mechanical behavior of tension and compression of AM Ti6Al4V varies, depending on the strain rate [62]. Rozumek and Hepner reported the influence of oxygenation time and heat treatment on fatigue crack propagation under bending in alloy Ti6Al4V [63,64]. Therefore, the fatigue strength of AM Ti6Al4V may differ, depending on the peening method and AM process, just as the strain rate during peening is dependent on the peening method.

In previous studies, the effects of the mechanical surface treatment of AM metals were reported using shot peening [5,6,13,65,66,67], ultrasonic peening [35,68], laser peening [13,66], and cavitation peening [13,65]. It should be noted that shock waves are used in submerged laser peening, as laser ablation and the collapse of the cavitation bubble, which is generated after laser ablation, producing shock waves [69,70,71]. In the case of cavitation peening, the impact induced by a micro jet and/or shock wave at the cavitation bubble collapse [72] is used for the mechanical surface treatment of metallic materials [70,73]. As shots are not used in cavitation peening, there is no material transfer due to shot collisions. This is a great advantage of cavitation peening for the mechanical surface treatment of medical implant applications. The impacts induced by cavitation peening and submerged laser peening might reach the bottom of deep valleys, which are produced by the AM process, and the impacts might enhance the fatigue properties of AM Ti6Al4V with a rough surface, such as as-built surfaces. Regarding the experimental investigation of plastic deformation induced by various peening methods [74], the ratio of the plastic deformation depth and the depth of the dent depend on the peening method. It was also reported that mechanical properties, such as roughness, residual stress, hardness, and crack propagation, differ remarkably, depending on the peening method, such as shot peening, laser peening, and cavitation peening [75,76,77]. As is well known, a peening method changes the surface characteristics, such as roughness, hardness, residual stress, and dislocation density, etc., of metallic materials, and the effect of peening also depends on the materials. Thus, a systematic and experimental investigation should be required to establish a suitable peening method to enhance the fatigue properties of AM Ti6Al4V.

In the present paper, to find out the effects on the fatigue properties of AM Ti6Al4V treated by various peening methods, Ti6Al4V manufactured by DMLS and EBM was treated by cavitation peening, laser peening, and shot peening, and the fatigue performance was evaluated by a plane bending fatigue test, comparing it with that of as-built specimens. In order to investigate the peening effects on the surface characteristics of Ti6Al4V manufactured by DMLS and EBM, the surface roughness, hardness, and residuals stress of the surface, with and without peening, were measured, and the surface was observed by a scanning electron microscope and a laser confocal microscope. An experimental formula to estimate the fatigue strength from the surface roughness, surface hardness, and residual stress was proposed, and it was verified.

## 2. Materials and Methods

### 2.1. Titanium Alloy Manufactured by DMLS and EBM

Figure 1 shows the shape of the specimens for a displacement-controlled plane bending fatigue tester. The specimens were manufactured by DMLS and EBM. The thickness of the specimen manufactured by DMLS was 2.6 ± 0.2 mm, considering the result of a previous report [13], due to the limitations of the displacement and the maximum moment of the tester. In the case of EBM, the thickness was 2 ± 0.2 mm, which was also found in a previous report [13]. The stacking direction of both DMLS and EBM specimens was in the width direction. After the DMLS and EBM process and then the heat treatment, the edges of all the specimens were rounded by hand using rubber whetstones of #80 and #180 to reduce the crack initiation from the edges; this was also conducted in a previous report [13]. To investigate mechanical polishing, the surface of the specimens manufactured by DMLS was grinded by rubber whetstones of #80 and #180, and it marked as “grinding” in the present paper.

The particle used in the DMLS process was made of Ti6Al4V, and its average diameter was about 40 µm. The power, beam diameter, and scanning speed of the laser of DMLS was 400 W, 200 µm, and 7 m/s, respectively. The stacking pitch was 60 µm. After the DMLS process, the specimen was annealed at 923 K for 3 h to reduce the residual stress in the specimen. After the annealing, the solution heat treatment was conducted at 1208 K under vacuum conditions for 105 min, then cooled in argon. After that, aging was conducted at 978 K under vacuum conditions for 2 h, followed by argon gas cooling.

The particle used in the EBM process was made of Ti6Al4V, and its average diameter was about 75 µm. The spot size of the electron beam for selective melting was 0.2 mm in diameter, and the stacking pitch was 90 µm, which was the same in a previous report [13]. After the EBM process, the solution heat treatment and aging were conducted in the same way as the DMLS specimens.

### 2.2. Cavitation Peening

Figure 2 shows a schematic diagram of the peening section of the cavitation peening system. In the present experiment, a submerged high-speed water jet was used to generate cavitation. The high-speed water jet was pressurized by a plunger pump, with a maximum working pressure of 30 MPa and maximum flow rate of 3 × 10^−2^ m^3^/min and injected into a chamber filled with water. The injection pressure was controlled by the rotational speed of the invertor motor of the plunger pump. When the high-speed water jet was injected into the chamber, cloud cavitation was generated inside the nozzle and/or in the shear layer around the jet. A submerged jet with a cavitation bubble is called a cavitating jet [70]. On the impinging surface, cloud cavitation became a ring vortex cavitation, then collapsed. Thus, the area treated by the fixed cavitating jet shows an annular shape [70], and the scanning cavitating jet can treat the surface uniformly [78]. In the present paper, the specimen, which was put in a recess to make a flat surface, was placed perpendicular to the jet in the chamber. The specimen was treated by scanning the nozzle, which moved horizontally at a constant speed *v*. The throat diameter of the nozzle was 2 mm. As the cavitator and the guide pipe can enhance the aggressive intensity of the cavitating jet [79], both of them were installed in the nozzle. The diameter of the cavitator was 3 mm, which was optimized in a previous study [79]. As the geometry of the outlet bore at the nozzle affects the aggressive intensity of the cavitating jet, the length *L* and the diameter *D* of the bore were optimized as *L* = 16 mm and *D* = 16 mm [80]. As the cavitating flow was separated at the upstream corner of the nozzle, the standoff distance *s* was defined by the distance from the upstream corner of the nozzle to the surface of the specimen. When the specimen is set too close to the nozzle, even though there is the cavitating jet, the specimen is treated by water jet peening, in which water columns peen the target. The classification map, in which cavitation peening and water jet peening were classified using the normalized standoff distance and cavitation number, was proposed [73]. The standoff distance chosen in the present experiment was *s* = 222 mm, which was the same in a previous report [13], and this was the cavitation peening condition.

The processing time per unit length *t_p_* is defined as follows.
(1)$$t_{p}=\frac{n}{v}$$
where *n* is the number of passes. In the present experiment, the processing time per unit length *t_p_* = 10 s/mm, which was the same in a previous study [13], was used to compare the cavitation peening effect on Ti6Al4V manufactured by DMLS and EBM.

### 2.3. Laser Peening

Figure 3 shows a schematic diagram of the peening section of the laser peening system. In the present experiment, a submerged laser peening system was used. In most cases, in conventional submerged laser peening, the second harmonic of the Nd:YAG laser (i.e., 532 nm in wave length), is used to mitigate the absorption of laser power due to water. As mentioned in the introduction, the impact at the collapse of the bubble, which is developed and collapses after laser ablation, is also affective for peening. In experimental studies of bubble dynamics, the fundamental harmonic of the Nd:YAG laser (i.e., 1064 nm) is used to harness the heat [81]. Soyama revealed that submerged laser peening, using a laser of 1064 nm in wave length, introduced compressive residual stress when the distances in air and water were optimized [82]. Thus, in this experiment, the wavelength of 1064 nm was used, which was the same in a previous report [13]. The pulse energy, beam diameter, pulse width, and repetition frequency of the used Q-switched Nd:YAG laser were 0.35 J, 6 mm, 6 ns, and 10 Hz, respectively. To minimize the damage to a glass chamber due to the laser, the laser was reflected by mirrors and expanded by a concave lens, then focused by convex lenses onto the specimen placed in the water-filled chamber, whose thickness was 3 mm. The length, width, and depth of the chamber were each 150 mm. The deionized water was fed into the chamber at about 5 × 10^−3^ m^3^/min to remove particles caused by the ablation. It should be noted that the water was degassed to minimize the cushion effect at the bubble collapse. The focal distance of the final convex lens was 100 mm. In order to minimize damage to the mirrors and lenses due to the reflection between the final convex lens and the chamber, the convex was placed as shown in Figure 3. In the present laser peening, the distances in air *s_a_* and in water *s_w_* were 84 mm and 19 mm, respectively, which was the same in a previous study [13]. In the present condition, the diameter of the laser spot on the specimen surface was about 0.8 mm. The specimen was placed on a stage, which was moved by linear stepping motors in the vertical and horizontal directions. The pulse density *d_L_* was controlled by the horizontal speed *v_s_* and the stepwise movement in the vertical direction *s_v_* using the stepping motors. As the repetition frequency was 10 Hz, *d_L_* was defined by Equation (2).
(2)$$d_{L}=\frac{10\;n}{v_{h}\;s_{v}}$$

In the present experiment, *d_L_* was set to 5 pulses/mm^2^, with *n* = 1, *v_h_* = 4.46 mm/s, and *s_v_* = 0.448 mm, which were the same in a previous study [13], to compare the laser peening effect on Ti6Al4V manufactured by DMLS and EBM.

### 2.4. Shot Peening

Figure 4 illustrates a schematic diagram of the peening head of the shot peening system. In the present experiment, a recirculating shot peening system accelerated by a water jet [83] was used for shot peening. Shots made of stainless-steel Japanese Industrial Standards JIS SUS440C were installed in the chamber, whose diameter was 54 mm, and accelerated and recirculated in the chamber by the water jet, with an injection pressure of 12 MPa. The number and the diameter of the shots were 500 and 3.2 mm, respectively. The water jet was injected into the chamber through three holes with a diameter of 0.8 mm. The standoff distance from the nozzle to the specimen surface was 50 mm. To avoid a loss of shots, the specimen was set in the recess. It should be noted that no compressive residual stress was introduced into the stainless steel without shots, so the water jet scarcely had any effect on the titanium alloy. The chamber was moved by a motor to treat the specimen surface uniformly. *t_p_* was defined by the scanning speed *v* and the number of scans *n*, as shown in Equation (1). In the present experiment, *t_p_* was set to 1 s/mm, which was the same in a previous study [13], to compare the shot peening effect on Ti6Al4V manufactured by DMLS and EBM.

### 2.5. Fatigue Test and Evaluation of Surface Characteristics

The fatigue properties of the specimens, with and without peening, were evaluated by a conventional Schenk-type displacement-controlled plane bending fatigue tester at *R* = −1. In the case of the Schenk-type displacement-controlled plane bending fatigue tester, the displacement was produced by the position of an eccentric wheel, and the applied stress was calculated from the bending moment, measured by a load cell [13]. The span length at a fixed point was 65 mm, as shown in Figure 1, and the test frequency was 12 Hz. The applied stress *σ_a_* at the test was calculated from the bending moment *M*, the width of the specimen *b* (i.e., 20 mm), and the thickness *δ* measured by a digital caliper, with an accuracy of 0.01 mm, as shown in Equation (3).
(3)$$\sigma_{a}=\frac{6\;M}{b\;\delta^{2}}$$

As mentioned in the introduction, the fatigue properties of AM Ti6Al4V are remarkably affected by the surface roughness. Thus, the arithmetic mean roughness *Ra* and the maximum height of the roughness *Rz* were measured by a stylus type profilometer. The used cutoff length was 2.5 mm, as the surface of the as-built AM Ti6Al4V was rough. To obtain the mean value and the standard deviation, the surface was measured three times in each case. As the bending stress was applied in the longitudinal direction in Figure 1, the surface roughness was measured in the longitudinal direction of the specimen.

In order to compare the work hardening effect of various peening methods with the as-built AM Ti6Al4V, the hardness was measured using a Rockwell superficial hardness tester and a 120 degree diamond conical indenter, with an initial load of 3 kgf (29 N) and an applied load of 15 kgf (147 N), which was the same in a previous study [13]. The measured hardness was identified by *H_R15N_*. The hardness was measured five times in each case to obtain the mean value and the standard deviation.

As one of the remarkable peening effects on the surface is the introduction of compressive residual stress, the residual stress *σ_R_* on the surface was measured by the 2D method [84] using an X-ray diffraction apparatus, with a two-dimensional detector, as the sin^2^
*ψ* − 2*θ* diagram of the laser peened metal was curved [85]. The optimal conditions for residual stress measurement using a 2D method, considering the error, were already reported [86]. The X-ray tube used was Cu Kα, and it was operated at 35 kV and 40 mA. The diameter of the collimator was 0.8 mm, and the diffraction from the 8 mm × 8 mm area on the surface was obtained by moving the specimen perpendicularly to the X-rays. The lattice plane (*h k l*) used for the measurement was the Ti (2 1 3) plane, and the diffraction angle, without strain, was 139.5 degrees. The depth for the 90% contribution to the diffracted X-rays was 12.4 µm in the present condition. The 24 diffraction rings from the specimen at various angles were detected, and the exposure time per frame at each single position was 5 min. As mentioned above, the bending stress was applied in the longitudinal direction of the specimen, and the residual stress in the longitudinal direction of the specimen are discussed in the present paper.

In order to observe the surface characteristics of Ti6Al4V manufactured by DMLS and EBM, the specimen surface was observed by a laser confocal microscope and a scanning electron microscope (SEM). To distinguish the crack initiation point, the fractured surface of the specimen was also observed by SEM.

## 3. Results

### 3.1. Surface Characteristics of Ti6Al4V Manufactured by DMLS and EBM

In order to investigate the effect of various peening methods on Ti6Al4V manufactured by DMLS and EBM, Figure 5 and Figure 6 show the aspects of specimens manufactured by DMLS and EBM, observed by a laser confocal microscope. In Figure 5 and Figure 6, the aspect and height data, which are revealed by the color map, from blue to red, were combined to clearly show the rough surface of AM Ti6Al4V. For both the as-built specimens manufactured by DMLS and EBM, a lot of un-melted particles are on the surface, and deep valleys, whose directions are perpendicular to the stacking direction, are shown. As the averaged diameter of the used particle for DMLS and EBM is 40 µm and 75 µm, respectively, the diameter particles on the DMLS surface are nearly half those on the surface of the EBM specimen. After cavitation peening and laser peening (see Figure 5b,c and Figure 6b,c), some particles were removed by impacts induced by the cavitation collapse and pulse laser. However, a lot of particles still exist on the surface. In the case of shot peening, the un-melted particles on the surface were deformed for both cases of DMLS and EBM, but deep valleys were still on the surface. As shown in Figure 5e, the deep valleys were not removed in the present grinding. In the case of cavitation peening after grinding, as shown in Figure 5f, the valleys are more remarkable. This is probably because the burrs generated by the grinding were cleaned by the cavitation impacts.

In order to compare the surface roughness, with and without peening, quantitatively, Figure 7a reveals the arithmetical mean roughness *Ra*, and Figure 7b shows the maximum height of the roughness profile *Rz*. The *Ra* and *Rz* of the as-built DMLS specimen are 11.7 ± 1.0 µm and 56.5 ± 2.7 µm, respectively and those of the EBM specimen are 19.3 ± 1.3 µm and 116.1 ± 9.9 µm, respectively. As shown in Figure 7, the surface of the EBM specimen is nearly two times rougher than that of the DMLS specimen, as the diameter of the used particles of the EBM specimen is about two times larger than that of the DMLS specimen. It should be noted that the *Rz* of both the DMLS and EBM specimens was 1.5 times larger than that of the diameter of the used particles. After cavitation peening, the *Rz* and *Ra* for both the DMLS and EBM specimens were slightly decreased, as some un-melted particles on the surface were removed by the cavitation impacts. On the other hand, the *Rz* and *Ra* of laser peening are nearly equivalent or slightly larger than those of the as-built specimen, even though some particles were removed from the surface by impacts induced by the laser ablation and the collapse of the bubble, which was generated after laser ablation. As mentioned in Section 2.3, the surface was treated in a 0.448 mm stepwise movement. Thus, a small wavy pattern in the 0.448 mm step was introduced by laser peening due to plastic deformation. In the case of shot peening, the *Rz* and *Ra* for both the DMLS and EBM specimens were drastically decreased by the deformation of the un-melted particles. The *Rz* and *Ra* of grinding are slightly better than those of shot peening. In the case of cavitation peening after grinding, *Rz* and *Ra* are increased, compared with those of grinding, as the burrs are removed, and the cavitation impacts produce plastic deformation pits.

In order to compare the hardness of the specimens, the Rockwell superficial hardness *H_R15N_* is shown in Figure 8. As the surface of Ti6Al4V is very rough, the scatter band of *H_R15N_* in each case is relatively large. The hardness of the as-built DMLS specimen is *H_R15N_* = 67.1 ± 7.0, and this is slightly harder than that of the EBM specimen (i.e., *H_R15N_* = 61.8 ± 6.3). The *H_R15N_* of the DMLS specimen produced by cavitation peening was about 10% larger than that of the as-built Ti6Al4V, and the increment for the EBM specimen by cavitation peening was 4%. In the case of laser peening, *H_R15N_* was decreased in the DMLS specimen, and it was increased in the EBM specimen. This might be caused by local melting and local plastic deformation due to the pulse laser. In the case of shot peening, *H_R15N_* was increased in both the DMLS and EBM specimens. It should be noted that *H_R15N_* was affected by the surface profile, such as the skewness *R_sk_* [13], and the *H_R15N_* at *R_sk_* < 0, as in the case of shot peening, was larger than that at *R_sk_* > 0, even though the hardness was equivalent. Thus, the increment on *H_R15N_* by shot peening was caused by the work hardening and deformation of the surface. In any case, the peening treatments caused a work hardening effect on both the DMLS and EBM surfaces.

As residual stress *σ_R_* is also important factor for improving the fatigue strength of metallic materials by peening methods, Figure 9 shows the residual stress on the surface of the DMLS and EBM specimens, with and without treatments. The *σ_R_* of the as-built DMLS and EBM specimens is nearly zero. After cavitation peening, grinding, and shot peening, compressive residual stress was introduced. In the case of laser peening, the *σ_R_* of the DMLS specimen was tension, and the *σ_R_* of EBM was compression. As laser peening produces ablation and plastic deformation at the same time, tensile residual stress was introduced by the ablation, and compression was introduced by the peening through the plastic deformation. When 70 µm of the surface of the laser-peened specimen was removed, the *σ_R_* of EBM was −187 ± 11 MPa. It should be noted that the depth for the 90% contribution in the residual stress measurement was about 12 µm, and the *Ra* of the as-built, cavitation peening and laser peening are similar. Thus, it can be concluded that the measured value of the residual stress was affected by the roughness, but all the treatments for the DMLS and EBM specimens introduced compressive residual stress into the surface (see Appendix A).

### 3.2. Fatigue Properties of Ti6Al4V Manufactured by DMLS and EBM

In order to reveal the fatigue properties of Ti6Al4V manufactured by DMLS and EBM, Figure 10 reveals the result of the plane bending fatigue test. The applied bending stress *σ_a_* in Figure 10 was calculated by Equation (3) using *δ*, which was measured by the calipers. When the fatigue life and fatigue strength of the as-built specimen manufactured by DMLS and EBM were compared, those of the DMLS specimen were better than those of the EBM specimen. In the case of the DMLS specimen, the fatigue life at *σ_a_* ≈ 450 MPa was improved by 10 times through cavitation peening after grinding, 5.9 times through cavitation peening, 4.7 times through shot peening, 3.5 times through laser peening, and 1.5 times through grinding, compared with the as-built specimen. The fatigue strength *σ_f_*_1_ at *N* = 10^7^ was calculated using Little’s method [87], as shown in Table 1. The improvement ratio *R*_1_ in Table 1 means the ratio of the fatigue strength, compared with the as-built specimen, for each DMLS and EBM specimen. The ratio *R*_2_ in Table 1 shows the ratio between the fatigue strength of DMLS and EBM for each treatment. As shown in Figure 10 and Table 1, in the case of the DMLS specimen, the fatigue strength at *N* = 10^7^ was improved by 1.97 times through cavitation peening, 1.93 times through laser peening, 1.92 times through shot peening, and 2.41 times through cavitation peening after grinding, compared with the as-built specimen. In the case of EBM, it was improved by 1.75 times through cavitation peening, 1.87 times through laser peening, and 1.95 times through shot peening. In sum, all treatments improved the fatigue strength, and the fatigue strength of the DMLS specimen was slightly better than that of the EBM specimen, as shown in Table 1.

### 3.3. Cracks of Ti6Al4V Manufactured by DMLS and EBM

In order to investigate the mechanism of the improvement of the fatigue properties by the treatments, Figure 11 and Figure 12 show the aspects of the specimen surface near the fracture, observed by a scanning electron microscope (SEM), and Figure 13 and Figure 14 reveal the fractured surface.

In Figure 11 and Figure 12, black arrows show the cracks that were produced by the fatigue, and white arrows show the cracks that existed before the fatigue test (see Appendix B). As shown in Figure 11a and Figure 12a, the cracks indicated by white arrows were observed on the surface of the as-built specimen, and they exist at the bottom of the valleys, which were orthogonal to the stacking direction. These cracks or steep valleys were produced during the AM process by layer-by-layer stacking. As shown in Figure 11b and Figure 12b, these steep valleys were opened by the cavitation attack. As the typical cavitation erosion pattern shows a jaggy pattern, the cavitation impact concentrates at the bottom of the dent. Thus, when the surface roughness was smoothed, the concentration of the cavitation impact was slightly reduced. This is a reason why the cracks at the bottom of the specimen treated by cavitation peening after grinding were fewer, compared with the specimen treated by cavitation peening without grinding. Thus, the fatigue strength of the specimen treated by cavitation peening after grinding was the best. As shown in Figure 11c and Figure 12c, the laser-peened surface was partially melted and oxidized on the surface. In the case of shot peening, the top of the un-melted particles was deformed, as shown in Figure 11d and Figure 12d. For both laser peening and shot peening, the cracks at the bottom of the valleys, indicated by white arrows, in the specimen, were similar to those of the as-built specimen. Moreover, the impacts produced by the pulse laser and shot collision did not open or close these cracks.

In Figure 11 and Figure 12, the direction of the applied bending stress was in the horizontal direction. Thus, the cracks produced by the fatigue, which are indicated by the black arrows, were developed in the vertical direction, as shown in Figure 11 and Figure 12. As is clearly shown in Figure 11b,d and Figure 12a–d, the cracks were initiated at the bottom of the valleys and developed. Then, it can be concluded that the weakest point of the Ti6Al4V manufactured by DMLS and EBM is where there are cracks at the bottom of the valleys, which were produced in the AM process.

In order to investigate the crack initiation point at the fatigue fracture, Figure 13 and Figure 14 show the aspect of the fractured surface of the specimen, observed by the scanning electron microscope. The crack initiation points are shown by arrows. As is clearly shown in Figure 13a,d–f and Figure 14a–d, the surface of the un-melted particles can be clearly observed at the crack initiation point. Thus, it can be said that the cracks were initiated at the bottom of the valleys, as mentioned above. In the present paper, these are called surface defects. When the depth of the surface defect of the as-built specimen *δ_d_* was measured by the fractured surface using SEM, it was 206 ± 43 µm for the DMLS specimen and 188 ± 22 µm for the EBM specimen. As shown in Figure 7b, the *Rz* of the as-built specimen was 56.5 ± 2.7 µm for the DMLS specimen and 116.1 ± 9.9 µm for the EBM specimen. Importantly, the *δ_d_* was remarkably larger than the *Rz* for both the DMLS and EBM specimens.

## 4. Discussion

### 4.1. Fatigue Strength Considering the Depth of the Surface Defect

The fatigue strength *σ_f_*_1_ in Table 1 was calculated with the thickness of the specimen, measured by the caliper as *δ* in Equation (3). In particular, the *δ* in Figure 15 was used as the *δ* in Equation (3). In the previous report, the core part *δ* and the surface roughness *Rz* were considered to calculate the bending stress, as shown Equation (4). Thus, *δ* was used as *δ* in Equation (3) [13]. In the present paper, the fatigue strength *σ_f_*_2_, considering *Rz*, is shown in Table 2. As the core part is thinner than the thickness of the specimen, the increase of *σ_f_*_2_, compared with *σ_f_*_1_, was 3–9% for the DMLS specimen and 8–43% for the EBM specimen.

(4)$$\delta_{2}=\delta_{1}-2Rz$$

As shown in Figure 13 and Figure 14, the depth of the surface defect *δ_d_* is much larger than *Rz*. This means that the actual core part is thinner than *δ*_2_. Then, the fatigue strength *σ_f_*_3_ was recalculated using *δ_3_*, which was defined by Equation (5), instead of *δ*_2_. *σ_f_*_3_ is shown in Table 2. It should be noted that *δ_d_* is 206 µm for the DMLS specimen and 188 µm for the EBM specimen in the present study.
(5)$$\delta_{3}=\delta_{1}-2\delta_{d}$$

The *σ_f_*_3_ of the DMLS specimen was 561 ± 12 MPa for cavitation peening after grinding and 497 ± 12 MPa for cavitation peening. Elsewhere [88], the fatigue strength at 10^7^ of the bulk material made of Ti6Al4V through heat treatment was 545 ± 10 MPa. The fatigue strength of the present Ti6Al4V manufactured by DMLS was about 50% of the bulk Ti6Al4V, and it was improved by up to 90% of the bulk Ti6Al4V by cavitation peening, and it came to have a nearly equivalent strength to the bulk Ti6Al4V.

### 4.2. Experimental Formula to Estimate Fatigue Strength Improved by Mechanical Surface Treatments

In order to evaluate the effect of the surface roughness, surface hardness, and residual stress on the improvement of the fatigue strength by mechanical surface treatment, the experimental formula to estimate the improved fatigue strength *σ_f__est_* by mechanical surface properties was proposed in Equation (6). In Equation (6), *σ_f_*_0_ is the fatigue strength of the reference condition. It was reported that the fatigue strength was decreased with the maximum surface roughness [39], equivalent crack length of the defect [43], and root area parameter [44]. Thus, the parameter of the surface roughness should be placed in the denominator, as shown in the second term of the right-side member of Equation (6). The hardness suggests the tendency of the increase of the yield stress of the surface by the mechanical surface treatments, and the hardness is expressed in Equation (6) as in the third term. As the residual stress affects the fatigue properties of AM Ti6Al6V [15], *σ_R_* is placed in the fourth term. Δ*Rz’* and Δ*H_RNT_’* are the differences in *Rz* and *H_RNT_* between the reference condition and the estimated condition, normalized by the reference condition, as shown in Equations (7) and (8). The Δ*σ_R_* is the difference in residual stress between the estimated condition and the reference condition. In particular, the second term of the right-side member of Equation (6) reveals the effect of surface roughness. The third term shows the effect of work hardening (i.e., an increase of hardness). The fourth term reveals the effect of residual stress. The second term and the third term were normalized by the reference values.
(6)$$\sigma_{f\;est}=\sigma_{f0}+a\frac{1}{\;\Delta Rz\prime \;}\sigma_{f0}+b\;\Delta H_{R15N}\prime \;\sigma_{f0}+c\;\Delta \sigma_{R}$$
(7)$$\Delta Rz^{\prime }=\frac{\;Rz_{0}-Rz\;}{Rz_{0}}$$
(8)$$\Delta H_{R15N}\prime =\frac{\;H_{R15N}-H_{R15N}{}_{0}\;}{\;H_{R15N}{}_{0}}$$
(9)$$\Delta \sigma_{R}=\sigma_{R}-\sigma_{R0}$$

Here, *a*, *b*, and *c* are the constants calculated using the least squares method, and they reveal the contributions of the surface roughness, surface hardness, and residual stress, to the improvement of fatigue strength. The obtained constants are shown in Table 3, and the relationship between the experimental fatigue strength *σ_f_*_1_, *σ_f_*_2_, and *σ_f_*_3_ and the estimated fatigue strength *σ_f_*_1 *est*_, *σ_f_*_2 *est*_, and *σ_f_*_3 *est*_ is shown in Figure 16. The error bars in Figure 16 were calculated from the standard deviation of the measured values (i.e., *σ_f_*_1_, *Rz*, *H_RNT_*, and *σ_R_*) using error analysis [89]. In Table 3 and Figure 16, the values of the as-built EBM specimen were used as the reference condition to estimate the fatigue strength. In the estimation, the residual stress of laser peening was −187 ± 11 MPa, as compressive residual stress was introduced in the near surface, although the surface was abraded by laser ablation. As shown in Figure 16, the relations between the experimental fatigue strength and the estimated fatigue strength for all three cases (i.e., *σ_f_*_1_ − *σ_f_*_1 *est*_, *σ_f_*_2_ − *σ_f_*_2 *est*_, and *σ_f_*_3_ − *σ_f_*_3 *est*_, are roughly on the line. These results show that the improved fatigue strength of the AM titanium alloy, enhanced by surface mechanical treatments, can be estimated from the fatigue strength of the as-built specimen by measuring the surface roughness, surface hardness, and surface residual stress of the treated one using Equation (6). Now, let us investigate the effect of the surface roughness, surface hardness, and surface residuals stress in the improvement of the fatigue strength. As mentioned above, although the surface roughness of the cavitation peened and laser peened specimens were scarcely changed, the fatigue strength was drastically improved by both peening methods. This means that the effect of the surface roughness was smaller than the others in the present study. This is why *a* is relatively smaller than *b* and *c*. When the absolute values of *a*, *b*, and *c* were compared, the absolute value of *b* was relatively larger than the others. In particular, the contribution of *H_R15N_* was relatively large, and the scatter band of *H_R15N_* was relatively large, as shown in Figure 8. Thus, the scatter band of the estimated fatigue strength was relatively large, as shown in Figure 16. As the compressive residual stress (i.e., negative value) enhances the fatigue strength, *c* is negative. As shown in Table 3, the absolute value of *c* was 0.34 for *σ_f_*_1_ − *σ_f_*_1 *est*_, 0.38 for *σ_f_*_2_ − *σ_f_*_2 *est*_, and 0.39 for *σ_f_*_3_ − *σ_f_*_3 *est*_. As the plane bending fatigue test was used in the present study, and the maximum bending stress was applied at the surface, the fatigue strength was improved by 34–39% of the introduced compressive residual stress. As shown in Figure 16, the laser peening points for both the DMLS and EBM specimens are slightly further than the others, as the measured work hardening was not so large, compared with the other treatments. The correlation coefficient between the experimental fatigue strength and the estimated value was 0.896 for *σ_f_*_1_ − *σ_f_*_1 *est*_, 0.766 for *σ_f_*_2_ − *σ_f_*_2 *est*_, and 0.823 for *σ_f_*_3_ − *σ_f_*_3 *est*_. As the number of the dataset was nine in the present study, the probability of a non-correlation is less than 0.1% for *σ_f_*_1_ − *σ_f_*_1 *est*_, 1.6% for *σ_f_*_2_ − *σ_f_*_2 *est*_, and 0.6% for *σ_f_*_3_ − *σ_f_*_3 *est*_. When the probability of a non-correlation is less than 1%, it can be concluded that the relationship is highly significant. Thus, it can be concluded that the relationship between the experimental fatigue strength and the estimated fatigue strength is highly significant. In particular, the improved fatigue strength treated by the present post-processing can be estimated by the surface roughness, surface hardness, and surface residual stress.

## 5. Conclusions

In order to establish post-processing for the improvement of the fatigue strength of the additive manufactured (AM) Ti6Al4V, Ti6Al4V manufactured by direct metal laser sintering (DMLS) and electron beam melting (EBM) was treated by cavitation peening, laser peening, and shot peening, and the fatigue properties were evaluated by the displacement-controlled plane bending fatigue test. To clarify the mechanism of the improvement of the fatigue strength by mechanical surface treatments, the surface mechanical properties were measured, and the experimental formula to estimate the improved fatigue strength from the mechanical surface properties was discussed to compare the contribution of each parameter. The results obtained can be summarized as follows:(1)Cavitation peening, laser peening, and shot peening can improve the fatigue strength of Ti6Al4V manufactured by DMLS and EBM. In the case of DMLS, the improvements in the fatigue strength at *N* = 10^7^, compared with that of the as-built specimen, were 97% for cavitation peening, 93% for laser peening, and 92% for shot peening in the present condition. In the case of EBM, they were 75% for cavitation peening, 87% for laser peening, and 95% for shot peening in the present condition.(2)The fatigue strength of Ti6Al4V manufactured by DMLS is slightly better than that manufactured by EBM, with and without peening. The difference in fatigue strength at *N* = 10^7^ between the DMLS and EBM specimens was +9% for the as-built specimen, +23% for the cavitation peening specimen, +13% for the laser peening specimen, and +8% for the shot peening specimen.(3)The fatigue strength of the Ti6Al4V manufactured by DMLS treated by cavitation peening was improved to the same level as that of wrought Ti6Al4V, when the depth of the surface defects was considered in the calculation of the fatigue strength. The fatigue strength at *N* = 10^7^ of the DMLS specimen was 497 MPa for cavitation peening and 561 MPa for cavitation peening after grinding the surface.(4)The improvement of the fatigue strength of AM Ti6Al4V caused by the treatments can be estimated using the surface roughness, surface hardness, and residual stress.

## Figures and Tables

**Figure 1 materials-13-02216-f001:**
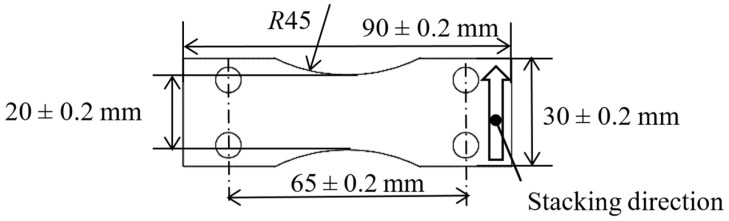
Geometry of the fatigue specimen for the displacement-controlled plane bending fatigue test. The thickness was 2.6 ± 0.2 mm for the direct metal laser sintering (DMLS) specimen and 2.0 ± 0.2 mm for the electron beam melting (EBM) specimen.

**Figure 2 materials-13-02216-f002:**
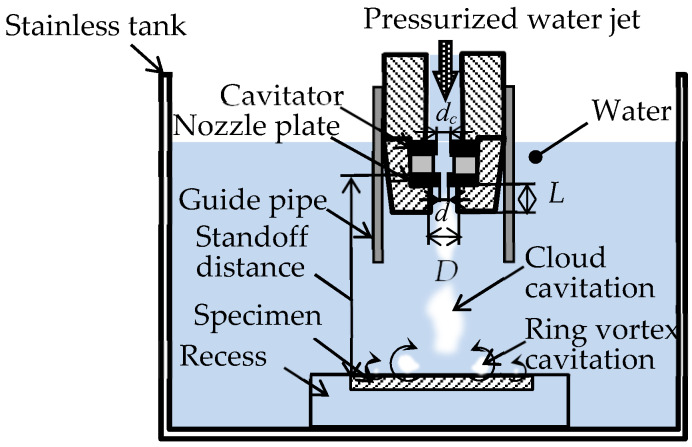
Schematic diagram of the peening section of the cavitation peening system.

**Figure 3 materials-13-02216-f003:**
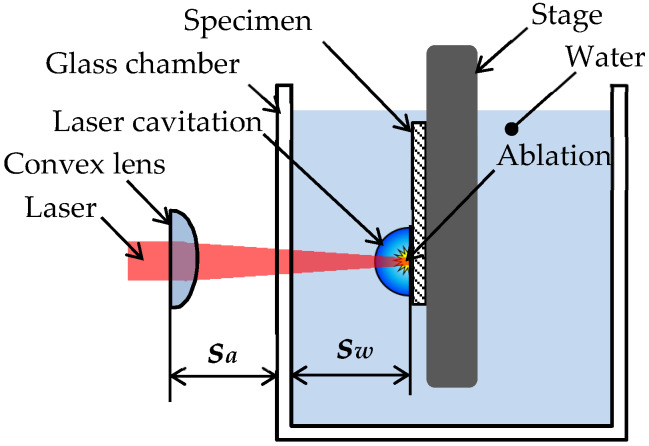
Schematic diagram of the peening section of the laser peening system.

**Figure 4 materials-13-02216-f004:**
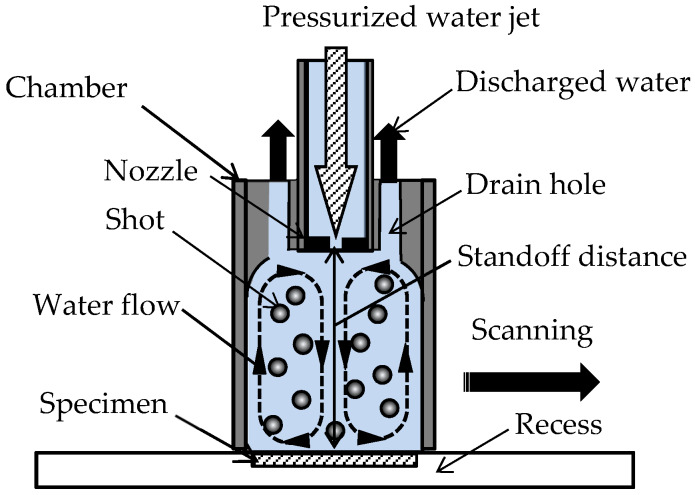
Schematic diagram of the peening head of the recirculating shot peening accelerated by a water jet.

**Figure 5 materials-13-02216-f005:**
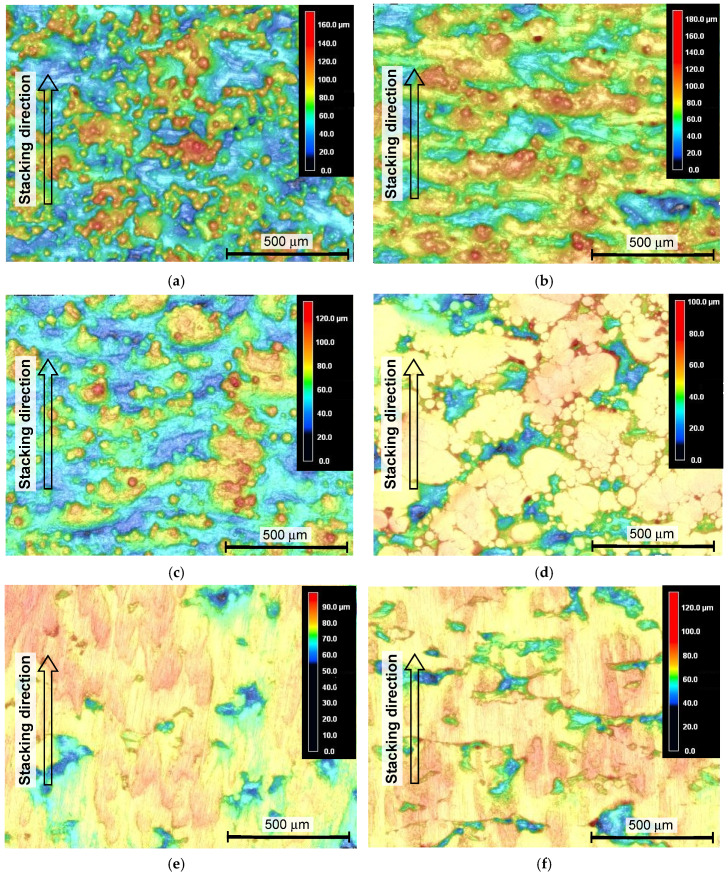
Aspects of the specimen manufactured by DMLS, observed by a laser confocal microscope. (**a**) As-built; (**b**) cavitation peening; (**c**) laser peening; (**d**) shot peening; (**e**) grinding; (**f**) grinding and cavitation peening.

**Figure 6 materials-13-02216-f006:**
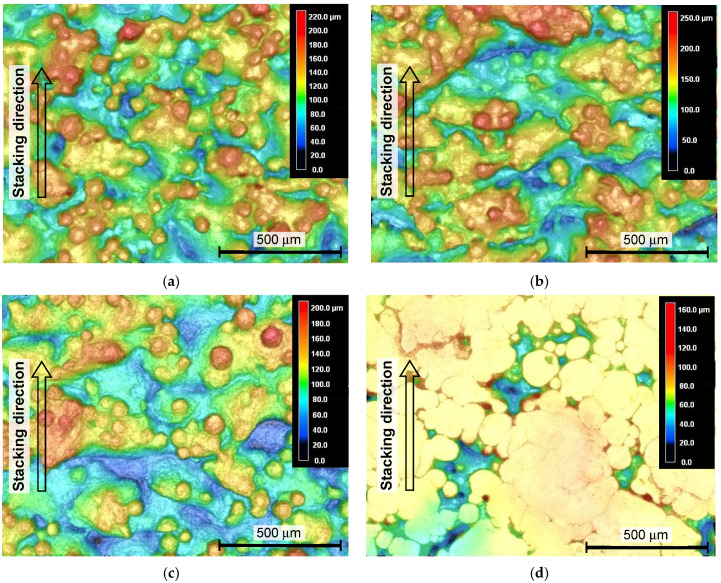
Aspects of the specimen manufactured by EBM, observed by a laser confocal microscope. (**a**) As-built; (**b**) cavitation peening; (**c**) laser peening; (**d**) shot peening.

**Figure 7 materials-13-02216-f007:**
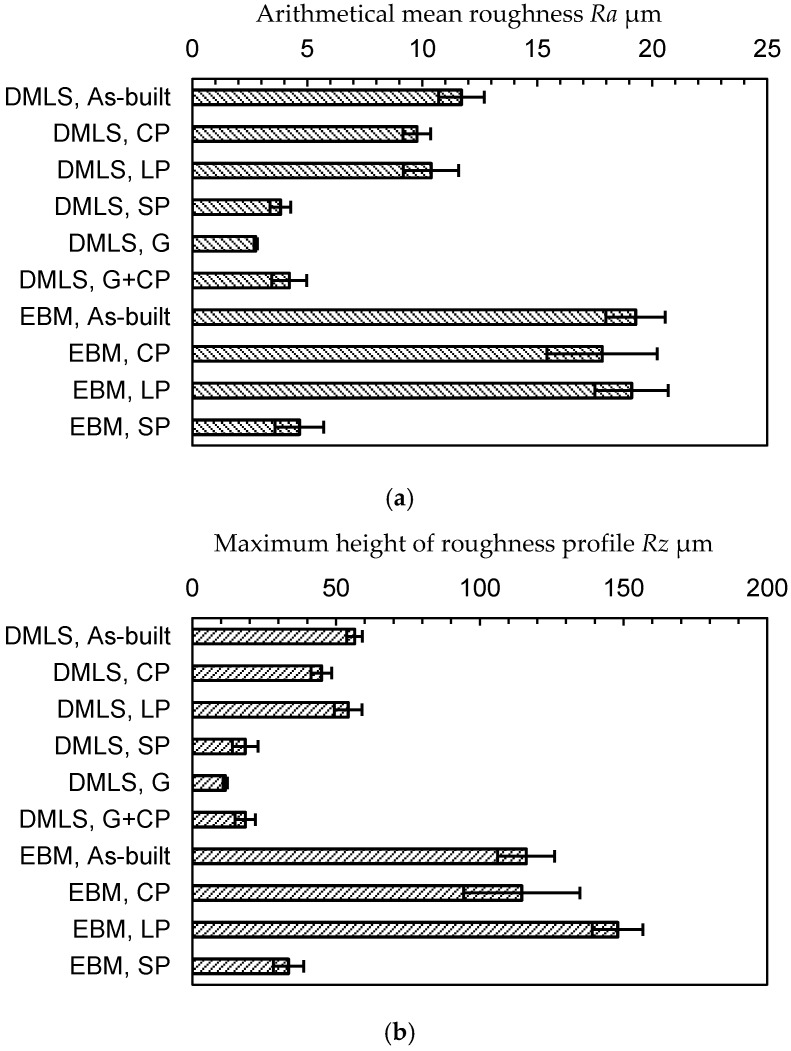
Surface roughness of the as-built Ti6Al4V manufactured by DMLS and EBM and the Ti6Al4V manufactured by DMLS and EBM through cavitation peening (CP), laser peening (LP), shot peeing (SP), grinding (G), and cavitation peening after grinding (G + CP). (**a**) Arithmetical mean roughness *Ra*. (**b**) Maximum height of the roughness profile *Rz.*

**Figure 8 materials-13-02216-f008:**
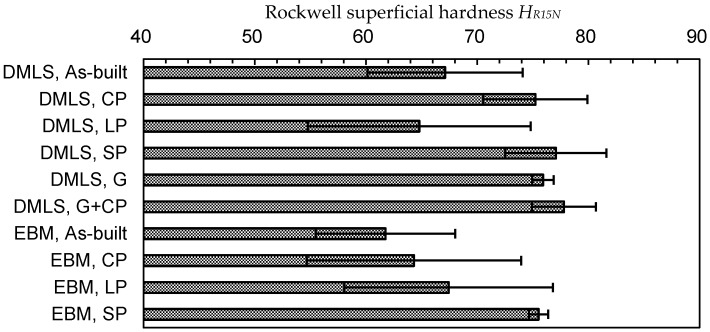
Surface hardness of the as-built Ti6Al4V manufactured by DMLS and EBM and the Ti6Al4V manufactured by DMLS and EBM through cavitation peening (CP), laser peening (LP), shot peeing (SP), grinding (G), and cavitation peening after grinding (G + CP).

**Figure 9 materials-13-02216-f009:**
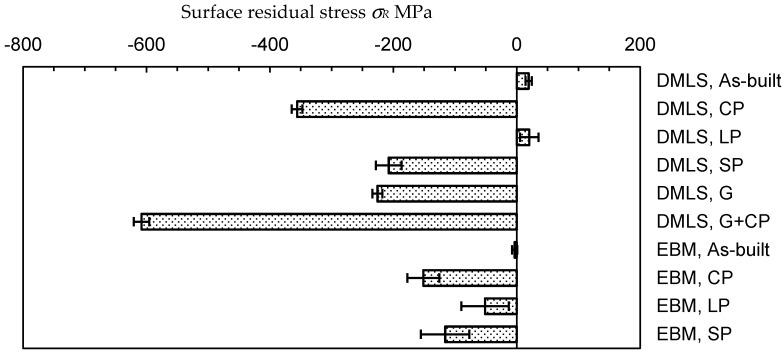
Surface residual stress of the as-built Ti6Al4V manufactured by DMLS and EBM and the Ti6Al4V manufactured by DMLS and EBM through cavitation peening (CP), laser peening (LP), shot peeing (SP), grinding (G), and cavitation peening after grinding (G + CP).

**Figure 10 materials-13-02216-f010:**
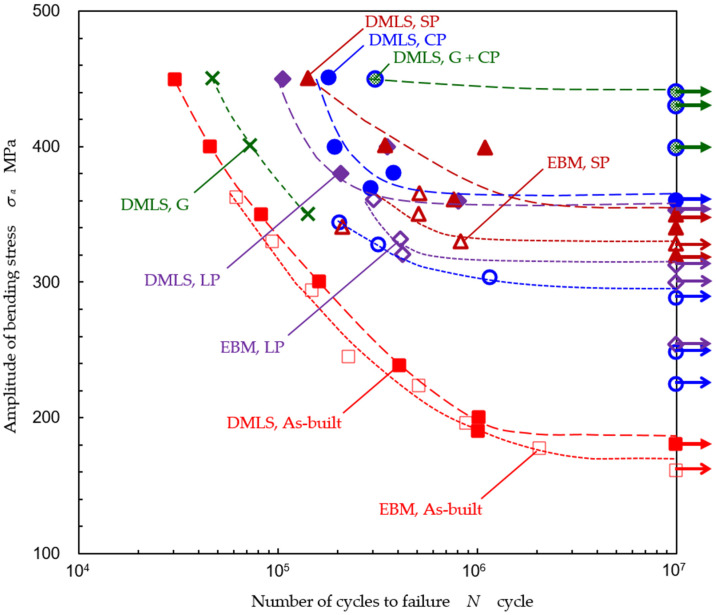
Improvement of the fatigue properties of Ti6Al4V manufactured by DMLS and EBM through cavitation peening (CP), laser peening (LP), shot peening (SP) grinding (G), and cavitation peening after grinding (G + CP), compared with the as-built specimen.

**Figure 11 materials-13-02216-f011:**
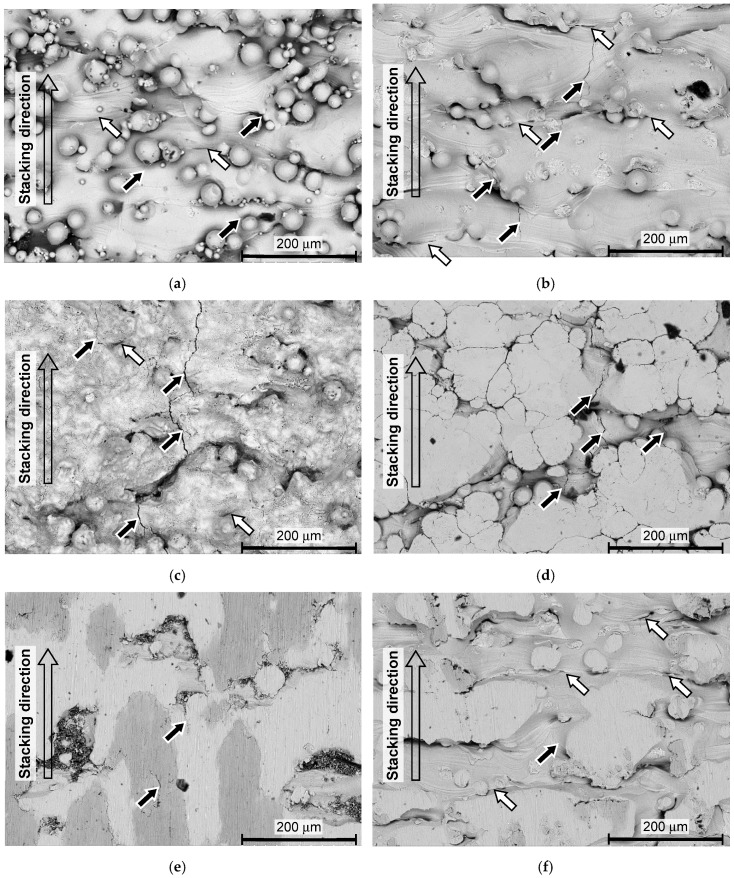
Aspects of the surface near the fracture of the specimen manufactured by DMLS, observed by scanning electron microscope (SEM). (**a**) As-built (*σ_a_* = 301 MPa, *N* = 162,400); (**b**) cavitation peening (*σ_a_* = 370 MPa, *N* = 291,600); (**c**) laser peening (*σ_a_* = 400 MPa, *N* = 352,200); (**d**) shot peening (*σ_a_* = 361 MPa, *N* = 764,800); (**e**) grinding (*σ_a_* = 350 MPa, *N* = 141,000); (**f**) grinding and cavitation peening (*σ_a_* = 450 MPa, *N* = 310,000).

**Figure 12 materials-13-02216-f012:**
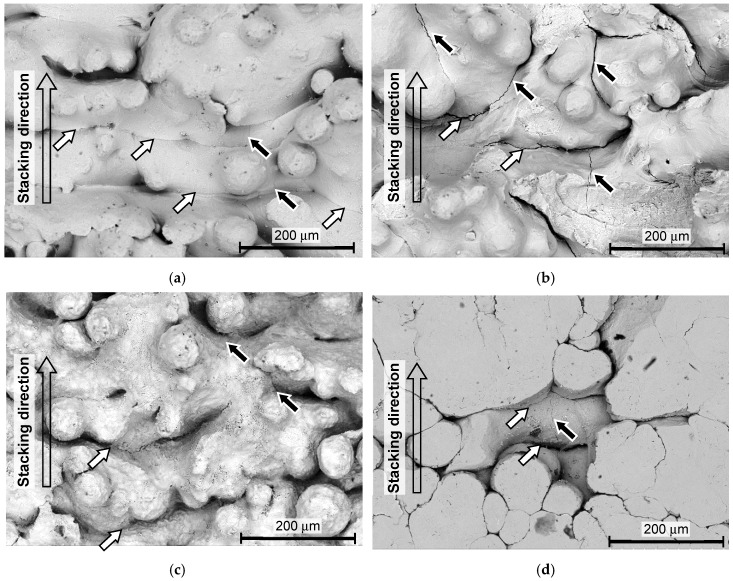
Aspect of the surface near the fracture of the specimen manufactured by EBM, observed by SEM. (**a**) As-built (*σ_a_* = 224 MPa, *N* = 510,300); (**b**) cavitation peening (*σ_a_* = 328 MPa, *N* = 319,300); (**c**) laser peening (*σ_a_* = 320 MPa, *N* = 420,900); (**d**) shot peening (*σ_a_* = 350 MPa, *N* = 505,700).

**Figure 13 materials-13-02216-f013:**
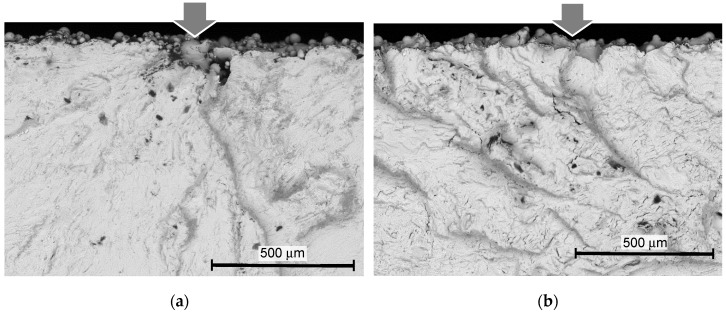
Aspects of the fractured surface of the specimen manufactured by DMLS, observed by SEM. (**a**) As-built (*σ_a_* = 301 MPa, *N* = 162,400); (**b**) cavitation peening (*σ_a_* = 370 MPa, *N* = 291,600); (**c**) laser peening (*σ_a_* = 400 MPa, *N* = 352,200); (**d**) shot peening (*σ_a_* = 361 MPa, *N* = 764,800); (**e**) grinding (*σ_a_* = 350 MPa, *N* = 141,000); (**f**) grinding and cavitation peening (*σ_a_* = 450 MPa, *N* = 310,000).

**Figure 14 materials-13-02216-f014:**
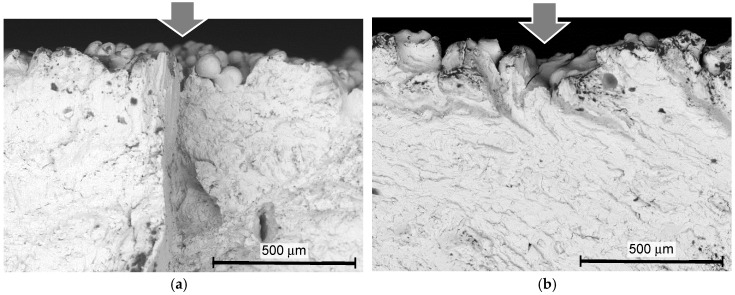
Aspects of the fractured surface of the specimen manufactured by EBM, observed by SEM. (**a**) As-built (*σ_a_* = 224 MPa, *N* = 510,300); (**b**) cavitation peening (*σ_a_* = 328 MPa, *N* = 319,300); (**c**) laser peening (*σ_a_* = 320 MPa, *N* = 420,900); (**d**) shot peening (*σ_a_* = 350 MPa, *N* = 505,700).

**Figure 15 materials-13-02216-f015:**
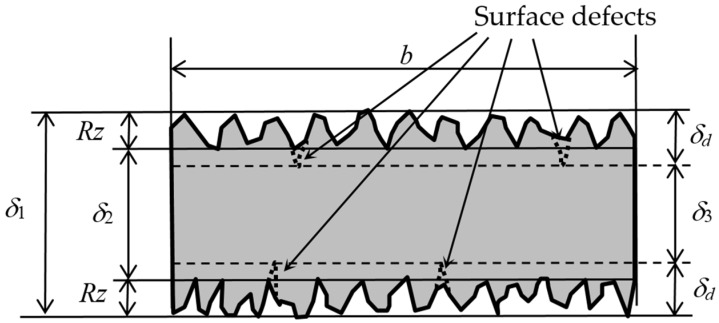
Schematic diagram of the thickness of the specimen for the calculation of the bending stress.

**Figure 16 materials-13-02216-f016:**
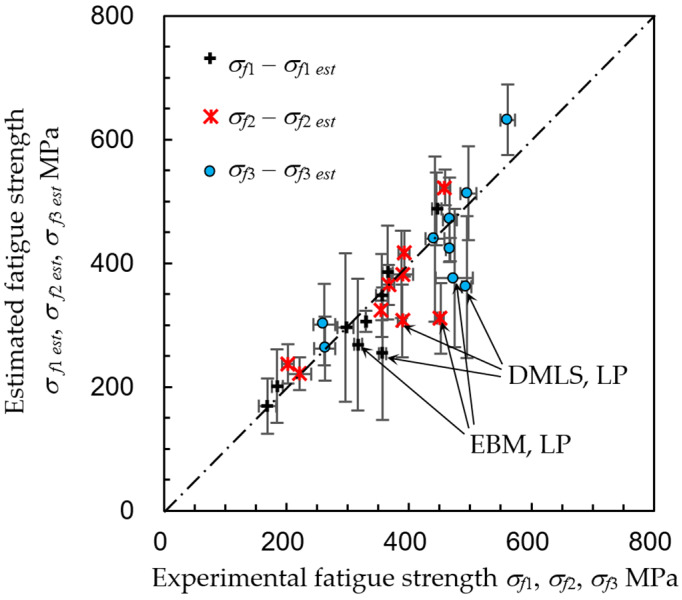
Relationship between the experimental fatigue strength and fatigue strength estimated from the surface roughness, surface hardness, and surface residual stress.

**Table 1 materials-13-02216-t001:** Fatigue strength and improvement ratio of the as-built Ti6Al4V manufactured by DMLS and EBM and the Ti6Al4V manufactured by DMLS and EBM through cavitation peening (CP), laser peening (LP), shot peeing (SP), grinding (G), and cavitation peening after grinding (G + CP).

	Number of Specimens	Fatigue Strength *σ_f_*_1_ MPa	Improvement Ratio *R*_1_	Ratio *R*_2_ = *σ_f_*_1 DMLS_/*σ_f_*_1 EBM_
DMLS, as-built	8	185 ± 9	1	1.09
DMLS, CP	5	365 ± 8	1.97	1.23
DMLS, LP	5	357 ± 6	1.93	1.13
DMLS, SP	7	355 ± 10	1.92	1.08
DMLS, G + CP	4	445 ± 8	2.41	—
EBM, as-built	8	169 ± 14	1	1
EBM, CP	6	296 ± 13	1.75	1
EBM, LP	6	317 ± 7	1.87	1
EBM, SP	5	329 ± 1	1.95	1

**Table 2 materials-13-02216-t002:** Fatigue strength considering the roughness and defect of the as-built Ti6Al4V manufactured by DMLS and EBM and the Ti6Al4V manufactured by DMLS and EBM through cavitation peening (CP), laser peening (LP), shot peeing (SP), grinding (G), and cavitation peening after grinding (G + CP).

	Fatigue Strength *σ_f_*_2_ MPa	Fatigue Strength *σ_f_*_3_ MPa
DMLS, as-built	202 ± 10	262 ± 18
DMLS, CP	392 ± 9	497 ± 13
DMLS, LP	389 ± 6	494 ± 8
DMLS, SP	366 ± 10	467 ± 12
DMLS, G + CP	459 ± 8	561 ± 12
EBM, as-built	221 ± 19	262 ± 17
EBM, CP	388 ± 18	442 ± 16
EBM, LP	451 ± 6	474 ± 30
EBM, SP	355 ± 1	468 ± 7

**Table 3 materials-13-02216-t003:** Constants to estimate the improved fatigue strength of the Ti6Al4V manufactured by DMLS and EBM through mechanical surface treatment using the surface roughness, surface hardness, and surface residual stress.

Constant	*σ_f_* _1 *est*_	*σ_f_* _2 *est*_	*σ_f_* _3 *est*_
*a*	0.01	0.01	0.01
*b*	2.59	1.17	1.95
*c*	−0.34	−0.38	−0.39

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
