# Peer review of "Effect of Various Peening Methods on the Fatigue Properties of Titanium Alloy Ti6Al4V Manufactured by Direct Metal Laser Sintering and Electron Beam Melting"

_materials, 2020, doi:10.3390/ma13102216_

Round 1

Reviewer 1 Report

The aim of the paper is to study the effect of cavitation peening, laser peening and shot peening methods on Fatigue Properties of Titanium Alloy Ti6Al4V Manufactured by Direct Metal Laser Sintering and Electron Beam Melting.

The study of the influence of the effect of Peening Methods in Titanium Alloy Ti6Al4V for additive manufacturing technologies is a subject nowadays interesting from the scientific and industrial point of view, however unfortunately the research developed by the authors presents certain limitations:

1.- The results presented by the authors pointed that Cavitation peening, laser peening and shot peening processes can improve fatigue strength of Ti6Al4V manufactured by DMLS and EBM technologies. However, the authors do not detail the number of test specimens used in each experimental test, nor do they indicate the accurate measurements of each specimen, the dimensional deviations from the theoretical dimensions, etc. The additive manufacturing process may present deviations regarding the precision of the measurements as well as internal manufacturing defects such as small pores, anisotropy, etc. The authors also do not detail the number of test specimens required by the standard.

2.- Unfortunately, the equations of the paper are not properly presented, which prevents their correct evaluation. Authors are encouraged to improve the presentation of the equations throughout the paper and of the figures in the paper starting on page 15.

3.- The authors present the results of the experimental tests on test specimens that greatly improve Fatigue Properties of Titanium Alloy Ti6Al4V, however, there may be great differences when applying the results obtained from test specimens on industrial parts manufactured with additive technology. For this reason, the authors are recommended to establish the limits of their research, as well as to add an experimental test under fatigue conditions on a real part manufactured by additive manufacturing and Titanium Alloy Ti6Al4V, which could validate the impact of the results obtained.

Author Response

Thank you for reviewing our manuscript and providing comments. We would like to respond to your comments and have revised the manuscript as attached file.

Reviewer 2 Report

The topic of fatigue of AM parts is very important nowadays, so this topic is of great interest. Major revisions are in order to improve the quality of the manuscript.

English is sub-standard and needs to be significantly improved.

The origin of residual stresses in AM processes should also be briefly discussed. Refer to 10.1016/j.pmatsci.2019.100590 and 10.1016/j.scriptamat.2018.04.024 and update the introduction accordingly.

“and EBM have different metallographic structures”: why? Please clarify.

“then the heat treatment”: what was the heat treatment conditions?

In fig 5 what is the color scale? Please add a scale for the readers to know the differences along the samples.

How was the error bar for the roughness determined?

How were the residual stresses determined? And the error bars? Can the authors provide an image of the 24 diffraction rings? This could be of interest to be better described in the experimental procedure.

“arrows show the cracks which existed before the fatigue test”: how do you know this?

Why can we see any striations in the fracture surfaces of the samples?

Pag 15 and 16 are not formatted properly. Please address this. The same for all the equations in the paper.

The discussion on how the graph of fig 16 was obtained needs a more in-depth discussion. It is very vague.

Also, do the authors have microscopy images of the as-built samples? This is important to show the original microstructures including defects.

Author Response

(The authors gave the same response as above.)

Reviewer 3 Report

Comments on the paper

  1. Equations (1), (2) and others has broken up.
  2. Page 6, lines 206-209 - please state the stress, displacement used during fatigue tests.
  3. What are the mechanical properties of the alloy tested?
  4. The captions of Figures 5, 6 and others should include (a), (b) ... etc.
  5. Fig. 16 clipped (half of the figure)?
  6. It would be worthwhile to quote in the introduction also papers of: 1) Rozumek D., Hepner M., Influence of microstructure on fatigue crack propagation under bending in the alloy Ti-6Al-4V after heat treatment. Mat.-wiss. u. Werkstofftech. 2015, 46, 1088-1095, 2) Rozumek D., Hepner M. Influence of oxygenation time on crack growth in titanium alloy under cyclic bending. Materials Science 2011, 47, 89-94.

Author Response

(The authors gave the same response as above.)

Round 2

Reviewer 1 Report

The authors have addressed satisfactorily the points raised during the review.
The article can now be published in its current version

Reviewer 2 Report

The manuscript was significantly improved and can now be accepted for publication.

Make sure that the formating of the images is correct. In my pdf they are scattered all over the place.